# Multi-Omics Integration for the Design of Novel Therapies and the Identification of Novel Biomarkers

**DOI:** 10.3390/proteomes11040034

**Published:** 2023-10-20

**Authors:** Tonci Ivanisevic, Raj N. Sewduth

**Affiliations:** VIB-KU Leuven Center for Cancer Biology (VIB), 3000 Leuven, Belgium; tonci.ivanisevic@kuleuven.be

**Keywords:** OMICs, data analysis, cancer, proteomics, data integration

## Abstract

Multi-omics is a cutting-edge approach that combines data from different biomolecular levels, such as DNA, RNA, proteins, metabolites, and epigenetic marks, to obtain a holistic view of how living systems work and interact. Multi-omics has been used for various purposes in biomedical research, such as identifying new diseases, discovering new drugs, personalizing treatments, and optimizing therapies. This review summarizes the latest progress and challenges of multi-omics for designing new treatments for human diseases, focusing on how to integrate and analyze multiple proteome data and examples of how to use multi-proteomics data to identify new drug targets. We also discussed the future directions and opportunities of multi-omics for developing innovative and effective therapies by deciphering proteome complexity.

## 1. Introduction

The complexity of biological systems is beyond the scope of single-omics studies, which only focus on one type of biological molecule. To fully understand the molecular mechanisms and interactions that underlie biological functions and diseases, it is necessary to integrate data from multiple omics levels, such as genomics, transcriptomics, proteomics, metabolomics, and epigenomics. This is the essence of multi-omics, an emerging approach that aims to provide a comprehensive and systematic view of biological systems. Multi-omics has been applied to various fields of biomedical research, such as diagnostics, drug discovery, personalized medicine, and synthetic biology. By combining different types of omics data, multi-omics can reveal novel insights into the molecular basis of diseases and drug responses, identify new biomarkers and therapeutic targets, and predict and optimize individualized treatments. Multi-omics has the potential to revolutionize the field of pharmaceutical sciences and enable the development of innovative and effective therapeutics.

However, the multi-omics approach faces many challenges, such as data heterogeneity, integration, analysis, interpretation, and validation. The high dimensionality, diversity, and complexity of multi-omics data pose significant computational and statistical difficulties for data integration and analysis [1]. The biological interpretation and validation of multi-omics results require extensive knowledge of the field of interest and experimental verification.

This review highlights the recent advances and challenges of multi-omics for the design of novel pharmaceutical therapies [2]. One of the significant challenges of drug discovery is integrating different omics data, such as genomics, transcriptomics, proteomics, and metabolomics, to identify and validate novel drug targets and biomarkers. Omics data can provide a comprehensive and holistic view of the molecular and cellular mechanisms of diseases, as well as the effects of drugs on various biological systems [3]. However, omics data are also complex, heterogeneous, and high-dimensional, requiring advanced computational methods and tools to analyze and interpret them.

The first step in multi-omics studies is to collect omics data from different sources or platforms. Depending on the research question and design, omics data can be obtained from different levels of biological organization (e.g., cell, tissue, organ), different types of samples (e.g., blood, urine, biopsy), different time points or conditions (e.g., before or after treatment), or different individuals or populations (e.g., healthy, diseased, in remission). Omics data can also be generated by different technologies or methods (e.g., RNA/DNA sequencing, mass spectrometry). The quality and quantity of omics data can vary greatly depending on the experimental design and/or procedures. The next step in multi-omics studies is to integrate omics data from different sources or platforms. Data integration aims to combine omics data in a meaningful way that preserves or enhances the information content of each dataset. Data integration can be challenging, depending on the type of omics data required to be combined for the analysis [4].

## 2. Main Text

### 2.1. Different Approaches for Multi-Omics Data Integration

There are different approaches and strategies for integrating omics data for drug discovery, depending on the type, quality, and availability of the data, as well as the biological question and hypothesis [5,6] (Figure 1).

Some of the common methods include:−Conceptual integration: This method involves using existing knowledge and databases to link different omics data based on shared concepts or entities, such as genes, proteins, pathways, or diseases. For example, one can use gene ontology (GO) terms or pathway databases to annotate and compare different omics data sets and identify common or specific biological functions or processes [7]. This method is useful for generating hypotheses and exploring associations between different omics data, but it may not capture the complexity and dynamics of the biological system. Open-source pipelines such as STATegra [8] or OmicsON [9] have recently demonstrated an enhanced capacity of the framework to detect specific features overlapping between the compared omics sets;−Statistical integration: This method involves using statistical techniques to combine or compare different omics data based on quantitative measures, such as correlation, regression, clustering, or classification [10]. For example, one can use correlation analysis to identify co-expressed genes or proteins across different omics data sets or use regression analysis to model the relationship between gene expression and drug response [11]. This method is useful for identifying patterns and trends in the omics data, but it may not account for the causal or mechanistic relationships between the omics data;−Model-based integration: This method involves using mathematical or computational models to simulate or predict the behavior of the biological system based on different omics data [12]. For example, one can use network models to represent the interactions between genes and proteins in different omics datasets or use pharmacokinetic/pharmacodynamic (PK/PD) models to describe the absorption, distribution, metabolism, and excretion (ADME) of drugs in different tissues or organs [13]. This method is useful for understanding the dynamics and regulation of the biological system, but it may require a lot of prior knowledge and assumptions about the system parameters and structure;−Networks and pathway data integration: This method involves using networks or pathways to represent the structure and function of the biological system based on different omics data. Networks are graphical representations of the nodes (e.g., genes, proteins) and interactions in the system, while pathways are collections of related biological processes or events that occur in a specific order or context [14]. For example, one can use protein–protein interaction (PPI) networks to visualize the physical interactions between proteins in different omics data sets or use metabolic pathways to illustrate the biochemical reactions involved in drug metabolism [15]. This method is useful for integrating multiple types of omics data at different levels of granularity and complexity, but it may not capture the temporal or spatial aspects of the system.

### 2.2. Aims of Multi-Omics Analyses

Data analysis aims to extract useful information or knowledge from omics data that can answer specific research questions or hypotheses (Figure 2). One of the main applications of multi-omics is to identify and validate new drug targets for various diseases. Drug targets are molecules that can be modulated by drugs to alter the disease state or phenotype. Drug targets can be proteins, genes, metabolites, or epigenetic marks that are involved in the pathogenesis or progression of diseases.

Multi-omics can help to discover and validate drug targets by:−Revealing the molecular signatures or profiles of diseases and drug responses using omics data from different levels of biological molecules [16]. For example, multi-omics can identify the genes, proteins, metabolites, and epigenetic marks that are differentially expressed or regulated in diseased versus healthy samples or individuals, or in responsive versus non-responsive samples or individuals to a given drug;−Constructing the molecular networks or pathways of diseases and drug responses using omics data from different levels of biological molecules [17]. For example, multi-omics can infer the interactions or relationships among genes, proteins, metabolites, and epigenetic marks that are involved in disease mechanisms or drug mechanisms of action;−Prioritizing the potential drug targets based on their relevance or importance to diseases and drug responses using omics data from different levels of biological molecules [18]. For example, multi-omics can rank genes, proteins, metabolites, and epigenetic marks based on their differential expression or regulation, network centrality, functional annotation, disease association, drug association, or other criteria;−Validating the selected drug targets using experimental methods or computational models that can test the effects of modulating the drug targets on diseases and drug responses. For example, multi-omics can provide guidance for designing experiments such as knockdowns, overexpressions, mutations, inhibitors, activators, or combinations thereof for the drug targets [19]. Alternatively, multi-omics can provide input for building computational models such as PK/PD models, systems pharmacology models, or machine learning models that can simulate the effects of modulating the drug targets [20].

Another main application of multi-omics is to predict and optimize drug responses for various diseases. Drug responses are the outcomes or phenotypes that result from administering drugs to treat diseases. Drug responses can be measured by various indicators such as efficacy, safety, toxicity, adverse effects, resistance, sensitivity, dosage, duration, frequency, or combinations thereof. Multi-omics can help to predict and optimize drug responses by:−Characterizing the inter-individual variability of drug responses using omics data from different levels of biological molecules [21]. For example, multi-omics can identify the genetic variants (e.g., single nucleotide polymorphisms (SNPs), copy number variations (CNVs), insertions/deletions (indels)), gene expression levels (e.g., mRNA levels), protein expression levels (e.g., protein levels), metabolite levels, and epigenetic modifications (e.g., DNA methylation levels) that influence how different individuals respond to a given drug;−Classifying the subtypes or groups of individuals with similar drug responses using omics data from different levels of biological molecules [22]. For example, multi-omics can cluster individuals based on their molecular signatures or profiles of drug responses into responders versus non-responders, sensitive versus resistant, or toxic versus non-toxic groups;−Predicting the optimal drug responses for individual patients using omics data from different levels of biological molecules [23]. For example, multi-omics can use machine learning methods such as SVMs, random forests, or neural networks to build predictive models that can estimate the efficacy, safety, toxicity, adverse effects, resistance, sensitivity, dosage, and duration of drug responses.

Some successful examples of multi-omics studies are listed in Table 1 and described below:−A study used multi-omics data from post-mortem brain samples to clarify the roles of risk-factor genes in complex diseases such as autism spectrum disorder (ASD) and Parkinson’s disease. The study integrated genomic, transcriptomic, epigenomic, and proteomic data to identify gene expression changes, DNA methylation patterns, and protein-protein interactions associated with ASD and Parkinson’s disease [24]. The study also revealed novel molecular pathways and potential therapeutic targets for these diseases;−A study that explained how to use multi-omics data from microbial metagenomes to investigate the interactions between plants, animals, and their microbiomes [25]. Another study integrated genomic, transcriptomic, proteomic, and metabolomic data from different host tissues and microbial communities to understand how the microbiome influences the host physiology, metabolism, immunity, and behavior [26];−Another example of multi-omics studies in cancer research is a work where authors used multi-omics data from tumor-infiltrating immune cells to develop a deep learning framework for predicting survival and drug response in breast cancer patients. Genomic, transcriptomic, proteomic, and epigenomic data were successfully integrated to identify the molecular signatures and profiles of immune cells in the tumor microenvironment [27];−Another example is a research article that describes the use of multi-omics in studying the molecular mechanisms and therapeutic targets of meningioma, a type of benign brain tumor. The authors used multi-omics data from human meningioma samples and cell lines to identify the functional roles of two genes, TRAF7 and KLF4, that are frequently mutated in meningioma [28]. The article demonstrates how multi-omics can provide novel insights into the molecular basis of diseases and drug responses, identify new biomarkers and therapeutic targets, predict and optimize individualized treatments, and design and engineer novel biological systems.

**Table 1 proteomes-11-00034-t001:** Example of multi-omics studies.

Title of the Article and Reference	Type of Data	Approach Used for Integration
An integrated multi-omics approach identifies epigenetic alterations associated with Alzheimer’s disease [24]	Transcriptomics, epigenomic, Chip-seq	GO analysis of genes, comparison to published data
Loss-of-function mutations in TRAF7 and KLF4 cooperatively activate RAS-like GTPase signaling and promote meningioma development [28]	Ubiquitome, proteome, interactome (ViroTrap) and transcriptome	Ingenuity Pathway analysis and network visualization using EnrichmentMap Cytoscape
Single-cell multi-omic integration compares and contrasts features of brain cell identity [29]	Single-cell RNA-seq and DNA methylation profiles	LIGER, an algorithm that delineates shared and dataset-specific features of cell identity
Multi-omics resolves a sharp disease-state shift between mild and moderate COVID-19 [30]	Proteome, single-cell Secretome (Isoplexis), Metabolome, single-cell RNA-seq	Cross-omic network analysis, and enrichment analysis using GSEA
Multi-omics delineation of cytokine-induced endothelial inflammatory states [31]	Secretome, proteome, phosphoproteome, transcriptome	Co-expression analysis was performed using the WGCNA, pathway analysis using clusterPofiler/WikiPathways
Multi-omics integration at single-cell resolution using bayesian networks: a case study in hepatocellular carcinoma [32]	Single-cell RNA-seq and copy number alterations	Bayesian networks
Spatial heterogeneity of infiltrating T cells in high-grade serous ovarian cancer revealed by multi-omics analysis [33]	Single-cell RNA-seq and whole genome sequencing, immunophenotyping (FACs), bulk RNA-seq analyses for immune cell infiltration	Gaussian Mixture Models
Computational integration of HSV-1 multi-omics data [34]	Ribosome profiling, RNA-seq, ATAC-seq	ContextMap2 which allows parallel mapping of RNA-seq reads against multiple genomes (host and microbial)
A “multi-omics” analysis of blood-brain barrier and synaptic dysfunction in APOE4 mice [35]	Single-nucleus RNA-sequencing, phosphoproteome proteome, interactome	Pathways analysis using FindMarkers, phosphorylated substrate to kinase network generation using Biogrid data
Multiomics signatures of type 1 diabetes with and without albuminuria [36]	Proteomics, lipidomics, metabolomics	Integration using MOFA, mapping using EggNog and KEGG databases

### 2.3. Different Types of Proteomics Data That Can Be Used for Multi-Omics Analyses

This review, as well as others, highlights the discrepancy between the interactome, proteome, and transcriptome [37]. The discrepancy between interactome/proteome and transcriptome is due to the difference between the levels of transcription of specific genes, translation of mRNA, and protein abundance or interaction in a biological system. This difference can be caused by various factors, such as post-transcriptional regulation, post-translational modification, protein degradation, protein–protein interaction, and environmental stimuli [38]. The discrepancy between interactome/proteome and transcriptome can have significant implications for understanding the molecular mechanisms and functions of biological systems, as well as for identifying potential biomarkers and therapeutic targets for diseases that are linked to protein complexity [39].

Proteome and phosphoproteome are two important concepts in the field of proteomics, which is the study of the entire set of proteins expressed by a cell, tissue, or organism under certain conditions. Proteome refers to the identity, expression levels, and modification of proteins, while phosphoproteome refers to the subset of proteins that are phosphorylated. This is a common post-translational modification that regulates protein function and signaling. Proteomics and phosphoproteomics can provide valuable information for the design of novel therapies [40], especially for diseases such as cancer, where protein expression and phosphorylation are often dysregulated. Table 2 comprises an updated list of the types of omics data publicly available for common cancer types.

Proteomics, ubiquitinome, and phosphoproteomics can also help to characterize molecular mechanisms and target modulators by integrating with other omics data, such as genomics, transcriptomics, and metabolomics. Proteomics can help identify potential biomarkers and protein expression patterns that can be used to assess disease prognosis, tumor classification, and identify potential responders for specific therapies [44]. Phosphoproteomics can help to understand cellular signaling and infer kinase activity, which is a key regulator of many cellular processes and a common target for drug development [45]. In line with these findings, a recent article described an overview of the online data publicly available in the field of cancer research, highlighting the discrepancy between different cancer types and potential multi-omics strategies [46].

The ubiquitinome refers to the set of proteins that are modified by covalently bound ubiquitin molecules, a small protein that regulates protein stability, localization, and function [47]. Ubiquitination is a reversible and dynamic process that can affect various cellular pathways such as the cell cycle, DNA repair, apoptosis, and autophagy. The ubiquitin system is involved in many diseases, such as cardiovascular diseases, cancer, neurodegeneration, inflammation, and infection [48]. Therefore, understanding and manipulating the ubiquitinome, which is the set of all ubiquitinated proteins in a cell or organism, could lead to new therapeutic strategies. One way to use the ubiquitinome for drug discovery is to identify biomarkers or signatures of the ubiquitinome that are associated with certain diseases or conditions. For example, changes in the levels or patterns of ubiquitination of certain proteins can indicate the presence or progression of a disease or the response or resistance to a treatment [49]. By measuring the ubiquitinome using proteomics or other methods, one can diagnose, monitor, or predict the clinical outcomes of patients. Moreover, one can use the information from the ubiquitinome to design novel therapies that target the underlying mechanisms or pathways of the disease, as in the case of Parkinson’s disease (PD) [50,51] or cardiovascular diseases such as Noonan syndrome [52].

Glycoproteome analysis is the study of the structure and function of proteins that are modified by glycans, which are complex carbohydrate chains attached to proteins [53]. Glycoproteins are involved in many biological processes and diseases, such as cell signaling, immune response, cancer, and viral infection. Glycoproteome analysis aims to identify glycoproteins, and their glycosylation sites. Glycosylation can affect the structure, stability, folding, interactions, and functions of proteins and thus regulate many cellular processes and pathways. Moreover, glycosylation can also influence the recognition and response of the immune system to foreign or abnormal cells, such as cancer cells or virus-infected cells. MS-based glycoproteome analysis can be performed using two complementary workflows: glycosylation site mapping and glycopeptide analysis. Glycosylation site mapping identifies the potential glycosylation sites that are occupied by glycans on the protein sequence, while glycopeptide analysis characterizes the specific glycan structures and compositions on each site. Glycoproteome analysis can be used as a powerful tool for disease diagnosis and therapy monitoring because glycosylation can serve as a biomarker that reflects the current status of the patient and the changes in the glycome due to disease progression or treatment [54]. Therefore, analyzing the glycome can reveal the alterations in glycosylation that are associated with different diseases, such as cancer, diabetes, Alzheimer’s disease, and infectious diseases. For example, cancer cells often have abnormal glycosylation patterns that affect their growth, invasion, metastasis, and immune evasion.

Protein acetylation is a type of post-translational modification that involves the addition of an acetyl group to a protein molecule. This modification can affect different amino acid residues of the protein, such as lysine, serine, and threonine. However, the most common and well-studied form of protein acetylation is the acetylation of lysine side chains. Protein acetylation can affect the structure, function, and interactions of proteins and regulate various biological processes such as metabolism and signaling. Protein acetylation can also occur at the N-terminus of the proteins, which is called N-terminal acetylation. N-terminal acetylation is catalyzed by a group of enzymes called N-terminal acetyltransferases (NATs). N-terminal acetylation can affect the protein’s lifetime by influencing its degradation, folding, localization, and interactions with other molecules. For example, N-terminal acetylation can protect proteins from being degraded by proteases that recognize unmodified N-termini, or it can target proteins for degradation by specific ubiquitin ligases. Acetylome analysis is the study of the global patterns and dynamics of protein acetylation using mass spectrometry and bioinformatics tools [55]. Acetylome analysis can reveal critical features of lysine acetylation, such as its abundance, distribution, conservation, and functional roles. Furthermore, it can reveal the changes in protein acetylation patterns and levels that are associated with disease pathogenesis and progression [56]. Acetylome can also be a target for therapy development, such as using drugs that modulate the activity of acetyltransferases or deacetylases, which are enzymes that add or remove acetyl groups from proteins. For example, elamipretide and nicotinamide mononucleotide are mitochondrial-targeted drugs that can restore the abundance and acetylation of proteins that are disrupted by aging in mouse hearts [57].

The interactome refers to the set of molecular interactions that occur in a particular cell. The term specifically refers to physical interactions among molecules, such as those among proteins, but can also describe sets of indirect interactions among genes, also called genetic interactions. Interactomes are generally displayed as graphs, where nodes represent molecules and interactions between the players [58]. Interactomes can help to understand the molecular mechanisms and functions of cells, as well as how they are affected by diseases or environmental changes. An example of using the interactome for drug discovery by identifying biomarkers or signatures of the interactome that are associated with certain diseases or conditions is the case of Alzheimer’s disease (AD) [59]. Interactomes of specific therapeutic targets can strengthen multi-omics analyses to identify regulatory mechanisms of proteins and potential therapeutic approaches [60].

One way to resolve the discrepancy between the interactome, proteome, and transcriptome is to use multi-omics approaches, which integrate data from different levels of biological molecules. For example, one study used multi-omics data from human brain organoids to identify the posttranscriptional regulation of ribosomal genes by a transcription factor called KLF4 [61]. Another study used multi-omics data from human melanoma samples to identify the molecular mechanisms and therapeutic targets of a novel oncogene called RREB1 [62]. While transcriptomics does not always correlate with protein levels and direct drug responses, the transcriptome can still be important for the design of therapies associated with large chromosomal rearrangements that can, for example, occur in cancer. Several papers have described the integration of RNA sequence, copy number aberration, and methylation to give a better understanding of cellular alterations associated with disease pathogenesis [63]. Such an approach can help identify genes that are affected by multiple types of genomic and epigenomic alterations. A recent article used transcriptome analysis to compare the effects of commonly observed chromosomic deletion on the gene expression in kidney epithelial cells and clear-cell renal cell carcinoma (ccRCC) samples [64].

## 3. Discussion

Multi-omics for drug discovery is a very exciting and promising field that aims to integrate and analyze data from different levels of biological molecules, such as DNA, RNA, proteins, and metabolites, to find new drugs and biomarkers for various diseases. Multi-omics can reveal novel insights into the molecular basis and mechanisms of diseases and drug responses, identify new therapeutic targets and pathways, predict and optimize individualized treatments, and design and engineer novel biological systems. However, multi-omics also faces many challenges, such as data heterogeneity, integration, analysis, interpretation, and validation. The development of advanced computational methods and tools, as well as the ethical implications of multi-omics data, need to be addressed and resolved. A recent piece of software, called Phenonaut 1.3, has also been designed to improve the auditability of multi-omics data integration [65], as several laboratory-developed tools are still not completely transparent. Multi-omics data integration has the potential to revolutionize the field of pharmaceutical sciences and enable the development of innovative and effective therapies, but it is still confronted with challenges such as auditability or consistency.

However, multi-omics data also poses some ethical challenges that need to be considered and addressed. Multi-omics data can reveal sensitive and personal information about individuals or groups, such as their genetic predispositions, health status, lifestyle, behavior, and environmental exposures. This information can be used for beneficial purposes, such as diagnosis, treatment, prevention, and research. However, it can also be misused or abused, such as through discrimination, stigmatization, exploitation, or coercion. Therefore, multi-omics data should be collected, stored, shared, and analyzed with respect for the privacy and confidentiality of the data subjects. This requires the implementation of appropriate technical and legal measures to protect the data from unauthorized access, use, or disclosure while at the same time balancing accessibility for researchers so as not to stifle scientific progress. It also requires the informed consent of the data subjects or their representatives to participate in multi-omics studies and to agree on the terms and conditions of data collection, storage, sharing, and analysis.

Multi-omics data can be affected by various sources of error, bias, or uncertainty that can compromise its quality and validity. These sources include the heterogeneity and complexity of biological systems, the variability and incompleteness of measurement methods, the inconsistency and incomparability of data formats and standards, the difficulty and subjectivity of data integration and interpretation, and the possibility and unpredictability of data changes over time. Therefore, multi-omics data should be generated, processed, reported, and evaluated with rigor and transparency to ensure its quality and validity. This requires the adoption of best practices and guidelines for multi-omics data generation, processing, reporting, and evaluation. It also requires the verification and validation of multi-omics data by independent methods or sources.

Multi-omics data can have significant scientific, clinical, social, and economic value for various stakeholders, such as researchers, clinicians, patients, communities, industries, governments, and society at large. However, there may be inequalities or disparities in access to and benefit from multi-omics data among different stakeholders due to various factors such as resource availability, technical capacity, legal regulation, ethical principles, or power relations. Therefore, multi-omics data should be shared and used in a fair and equitable manner that respects the rights and interests of all stakeholders. This requires the establishment of policies and mechanisms for multi-omics data sharing and use that balance the needs and expectations of different stakeholders. It also requires the recognition and reward of the contributions and efforts of different stakeholders in generating or using multi-omics data.

## 4. Conclusions

By analyzing data from various omics levels, such as the genome, transcriptome, proteome, metabolome, and microbiome, multi-omics integration can explore complex biological systems and find new biomarkers and therapeutic targets for different diseases, especially cancer. Multi-omics integration can reveal how different molecular entities and biological processes interact and relate to each other. Proteome complexity can make therapy design complicated due to the diversity and variability of protein forms and functions that are influenced by gene expression, alternative splicing, post-translational modifications, protein-protein interactions, and protein degradation. Proteome complexity poses a significant challenge for proteomics studies, as it demands advanced techniques and methods to detect and measure the different protein species in a specific biological system. Here we showed that the phosphoproteome, glycoproteome, acetylome, and ubiquitinome are important for such analyses in combination with the classical proteome or transcriptome.

One of the main objectives of multi-omics integration for the design of novel therapies and the identification of novel biomarkers is to address proteome complexity and understand how it impacts disease mechanisms and outcomes. For instance, some studies have used multi-omics integration to find new prognostic biomarkers by integrating multi-omics data from cancer patients to create an interactive web application for multi-omics data exploration and integration. They have also used it to apply proteomics in cancer research to identify molecular signatures and mechanisms related to tumor growth and metastasis. These studies have used various techniques and tools, such as mass spectrometry, network analysis, dimensionality reduction, machine learning, and bioinformatics pipelines, to analyze and integrate multi-omics data and address proteome complexity.

Multi-omics integration for the design of novel therapies and the identification of novel biomarkers is a promising and rapidly evolving field that can provide valuable insights into the molecular basis of diseases and potential interventions. However, it also faces some challenges and limitations, such as data quality, standardization, reproducibility, interpretation, and validation. Therefore, further research and development are needed to improve the methods and applications of multi-omics integration and to address proteome complexity comprehensively and reliably. Furthermore, data integration for multi-omics studies is a complex and challenging task that requires careful planning and execution of the study design, data generation, data processing, data analysis, and data interpretation. It also requires the development and application of appropriate methods and tools that can integrate multi-omics data robustly and reliably. Recent studies cover specific ways of data interpretation, especially for handling large datasets [66] and looking at predictive modeling [67]. By doing so, these studies provide a path to leverage the integrated connections of multi-omics data to gain a deeper understanding of biological systems and diseases.

## Figures and Tables

**Figure 1 proteomes-11-00034-f001:**
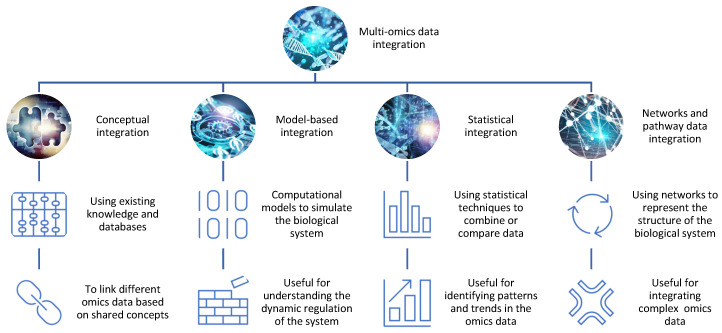
The integration of different omics data for drug discovery. There are different methods and tools for integrating omics data, such as conceptual integration, statistical integration, model-based integration, and network-based integration. Each method has its own advantages and limitations and can reveal distinct aspects of the biological system.

**Figure 2 proteomes-11-00034-f002:**
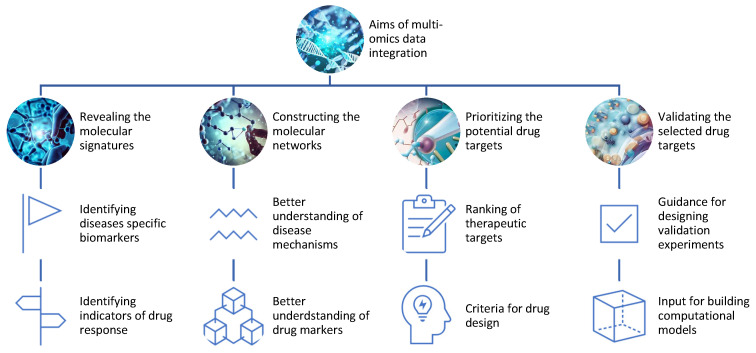
Aims of the integration of different omics data for drug discovery.

**Table 2 proteomes-11-00034-t002:** Publicly available omics data in the cancer research field in publicly available databases such as the Catalogue Of Somatic Mutations In Cancer (COSMIC), The Cancer Genome Atlas (TCGA) and Clinical Proteomic Tumor Analysis Consortium (CPTAC).

Cancer Type	Genome [41]	Transcriptome [42]	Methylome	Proteome [43]	Other CPTAC Data [43]
Acute Myeloid Leukemia	COSMIC	TCGA	TCGA	-	
Adrenocortical Carcinoma	COSMIC	TCGA	TCGA	-	
Bladder Carcinoma	COSMIC	TCGA	-	-	
Breast Carcinoma	COSMIC	TCGA	TCGA	CPTAC	Acetylome
Cervical Carcinoma	COSMIC	TCGA	TCGA	-	
Cholangiocarcinoma	COSMIC	TCGA	-	-	
Colorectal Adenocarcinoma	COSMIC	TCGA	-	CPTAC	
Esophageal Carcinoma	COSMIC	TCGA	-	-	
Gastric Adenocarcinoma	COSMIC	TCGA	-	-	
Glioblastoma	COSMIC	TCGA	-	CPTAC	Acetylome, Phosphoproteome, Proteome
Head and Neck Squamous Cell Carcinoma	COSMIC	TCGA	-	CPTAC	Phosphoproteome, Proteome
Hepatocellular Carcinoma	COSMIC	TCGA	-	-	Phosphoproteome, Proteome
Chromophobe Renal Cell Carcinoma	COSMIC	TCGA	-	-	
Clear Cell Renal Cell Carcinoma	COSMIC	TCGA	TCGA	-	
Papillary Renal Cell Carcinoma	COSMIC	TCGA	-	-	
Lung Adenocarcinoma	COSMIC	TCGA	TCGA	CPTAC	Phosphoproteome, Acetylome
Lung Squamous Cell Carcinoma	COSMIC	TCGA	TCGA	CPTAC	Ubiquitinome, Phosphoproteome
Mesothelioma	COSMIC	TCGA	-	-	
Ovarian Serous Adenocarcinoma	COSMIC	TCGA	TCGA	CPTAC	Glycoproteome, Phosphoproteome, Proteome
Pancreatic Ductal Adenocarcinoma	COSMIC	TCGA	-	CPTAC	
Paraganglioma and Pheochromocytoma	COSMIC	TCGA	-	-	
Prostate Adenocarcinoma	COSMIC	TCGA	-	-	
Sarcoma	COSMIC	TCGA	-	-	
Skin Cutaneous Melanoma	COSMIC	TCGA	-	-	
Testicular Germ Cell Cancer	COSMIC	TCGA	-	-	
Thymoma	COSMIC	TCGA	-	-	
Thyroid Papillary Carcinoma	COSMIC	TCGA	-	-	
Uterine Carcinosarcoma	COSMIC	TCGA	-	-	
Uterine Endo-metrioid Carcinoma	COSMIC	TCGA	TCGA	-	-
Uveal Melanoma	COSMIC	TCGA			

## Data Availability

Not applicable.

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
