# Peer review of "Multi-Omics Integration for the Design of Novel Therapies and the Identification of Novel Biomarkers"

_proteomes, 2023, doi:10.3390/proteomes11040034_

Round 1

Reviewer 1 Report

Several multi-omics reviews have been published recently. However, each has a different focus. This review has 65 references, half of them from the last two years, representing well the exponential growing of this field.

 Suggestions, questions:

Line 57                  Delete!

Fig. 2                      Improve the quality of the figure!

Table 1                  What do 28[29]-like references mean?

Table 2                  Please solve the COSMIC, TCGA, CPTAC abbreviations!

Line 242                I suggest adding some references to glycoanalysis, concerning both for glycan and modification site determination.

Line 247                Protein acetylation can also occur at the N- terminus of the proteins not only at the Lys sidechain. This PTM is related to the protein lifetime.

Line 297                What are the ethical implications of multi-omics data?

Author Response

We thank the reviewer for these very relevant comments. We have corrected the minor points, and expanded the discussion including ethical implications of multi-omics data. We have also added more information to the glycoanalysis and acetylation part of the main text.

We cannot delete line 57, as this title corresponds to the guidelines of the journal.

The figure 2 was replaced for better quality.

The references in Table 1 correspond to the articles described in each line. We have reformatted to make it more obvious.

The abbreviations in Table 2 were defined in the table legend.

We added information on glycoanalysis, currently line 240.

We added information on protein acetylation, currently line 257 to 270.

We have added a paragraph focusing on ethical implications of multi OMICs, line 320 to 332.

Reviewer 2 Report

Overview of the article:

·       The article titled “Multi-omics integration for the design of novel therapies and the identification of novel biomarkers” by the authors Ivanisevic T., and Sewduth, R. provides a review of the potentials of multi-omics integration and its applications in drug discovery and therapy design.

·       The article highlights how the concept of multi-omics, including, but not limited to, genomics, transcriptomics, and proteomics, can be utilized to gain a comprehensive understanding of biological systems. Its major focus is on the potential applications in drug discovery and personalized medicine, and touches base on the potential use for tracking disease mechanisms and therapeutic targets. The authors also introduce some of the major challenges derived from the complexity of the proteome, mentioning ethical aspects as well as data integration and analysis.

Strengths of the article:

·       The authors provide a comprehensive review and explain in detail the need for integrating data from the multi-omics areas to obtain a rounded understanding of biological systems.

·       The authors provide and reference examples of successful multi-omics applications to strengthen their main objective. Results from presented studies clearly demonstrate the utility of multi-omics approaches and use for potential therapeutic targets.

·       The authors provide detailed information on the complexity of the proteome, and the topic is revisited throughout the article. Some of the issues resulting from this complexity addressed are challenges of the protein diversity, PTMs and analyses execution and interpretation.

·       The authors provide overview of the challenges for data analysis and interpretation of experimental and predictive data. They provide references to several different approaches with the overall goal to emphasize the complexity of such determination.

·       The authors cover areas of very broad interest for biomarker and therapeutic target identification, which can be used to provide relevant biological insight potentially in health and disease.

Weaknesses of the article:

·       The article could benefit from a critical review and a more concise structure. The authors tend to repeat statements multiple times throughout. In addition, the authors tend to use more of a descriptive language, and it is recommended that they revise it to be more precise and on point. Readers may “be lost” throughout the long descriptive paragraphs and may find it challenging to get to point.

·       The authors provide a vast amount of information in the article, however they lack to provide their take on the specific topic, whether it is the complexity, the ethical concerns, or the limitations and challenges. The authors are encouraged to provide more critical analysis, especially discuss the potential impact of the challenges for example, on the research design, outcomes, and analysis.

·       One of the biggest challenges in this era of multi-omics approaches is the complexity of data integration. Even thought the authors tough base on this topic, it seems that they are not capturing sufficient details of the challenges in the area. The article could benefit more references for this part (there are multiple and more recent studies available which cover specific ways of data interpretation, especially for handling large datasets and looking at predictive modeling.

·       Another challenge the authors are reviewing in this article is the ethical aspects and potential challenges with regulations and considerations revolving around patient data and background information. The authors are encouraged to include few sentences about the current state of the ethics/regulation and elaborate what potential changes would need to be made to enable prospective multi-omics patient profiling or research studies.

·       Lastly, the authors are encouraged to review the Discussion paragraph and potentially revise it to be less repetitive, less descriptive, and more on point to capture the major take home points in the article. This paragraph might benefit from couple strong statements from the authors backed up by recent references focusing on the future steps and prospects of the multi-omics approaches elaborated in the article.

Please reference to the weaknesses section in the overall comments.

Author Response

We thank the reviewer for these very relevant comments and interest in our review. We have corrected the different points, and expanded the discussion including ethical implications of multi-omics data. We have also added one paragraph focusing on the challenges in data integration, citing references recommended by the reviewer.

We expanded the discussion including ethical implications of multi-omics data, currently lines 321 to 333.

We expanded the discussion with a paragraph on limitations of multi omics studies, lines 343 to 353.

We have also added one paragraph focusing on the challenges in data integration, citing references recommended by the reviewer, currently lines 383 to 393.

We have also clarified some parts of the main text, that could be seen as incomplete, lines 249 to 269.